# SPEED (SPlEnic Embolisation Decisions) study —Decision to treat acute traumatic splenic artery injury in the context of trauma protocol

Paul Jenkins[1], Jim Zhong[2,3]*, James Harding[4], Lexy Sorrell[5], Jason Smith[1], Victoria Allgar[5], Carl Roobottom[1]

1 Department of Diagnostic and Interventional Radiology, University Hospitals Plymouth NHS Trust, Plymouth, United Kingdom, 2 Department of Diagnostic and Interventional Radiology, Leeds Teaching Hospitals NHS Trust, Leeds, United Kingdom, 3 School of Medicine, University of Leeds, Leeds, United Kingdom, 4 University Hospital Coventry and Warrick NHS Trust, Coventry, United Kingdom, 5 Department of Statistics, University of Plymouth, Plymouth, United Kingdom

* jim.zhong@nhs.net

**Data Availability Statement:** No datasets were generated or analysed during the current study. All relevant data from this study will be made available upon study completion.

## Abstract

### Background

The spleen is commonly injured in trauma and this may be managed with a conservative approach, embolisation or splenectomy. There is uncertainty how splenic embolisation fits into the treatment paradigm and the delivery of IR services remains variable.

### Aims and objectives

The primary objectives are to determine if service design significantly affects splenic embolisation (SE) rates in AAST grade 2–5 acute traumatic splenic injuries (ATSI) across the Major Trauma Centres (MTCs) in England and to determine if variation in treatment affects SE outcomes in ATSI.

### Methods

We will include 5 years of data from traumatic splenic injury patients in the MTCs from 01/01/2016 to 31/12/2020 available from the Trauma Audit and Research Network (TARN) database. Inclusion Criteria will be all patients with ATSI registered with TARN. Those without a CT available to grade radiologically will be excluded. Data available from the TARN database and then correlated with data that will be collected at each MTC, where detail as to the embolisation technique, specific injury pattern, imaging based follow up and patient survival will be available. A short service-based questionnaire will be sent to each centre to establish centre-specific details such as on call rota, IR response activation, reporting practices and capture data around routine decision-making at that site. Data will be collected on an anonymised (REDCap) database. This project will evaluate the impact of service design on embolisation rates and outcomes, as well as evaluating the impact of the variation upon

**Funding:** This study has received funding from the British Society of Interventional Radiology Research Grant however the funders had no role in study design, data collection and analysis, decision to publish, or preparation of the manuscript.

**Competing interests:** The authors have declared that no competing interests exist.

treatment selection and outcomes. Logistic regression will be used to identify factors associated with treatment selection and mortality at 30 days.

# 1. Introduction

LAY SUMMARY: The spleen is often injured when the body sustains trauma. This leads to internal bleeding. The bleeding can be stopped by a big operation cutting open the belly or through a tiny hole in the groin where a small tube can be passed into a blood vessel and up to the spleen to stop the bleeding from within the blood vessels (called embolisation) using a variety of small metal plugs, or other materials. We do not know what types of injuries and which patients may benefit most from this treatment or the best way to perform this procedure and what material to use. Using a national database we hope to shed further light on this issue which can also be used to design further studies to better understand how the spleen can be best treated in trauma. The SPEED study summary flow chart is shown in Fig 1. See Table 1 for a study summary.

## 1.1. Background

Splenic embolisation (SE) is a minimally invasive procedure whereby the main or branches of the splenic artery are blocked to stop bleeding from the spleen. This is typically undertaken in the context of acute traumatic splenic injury, which is commonly diagnosed using a Computed Tomography (CT) scan by a diagnostic radiologist. The vessel can be accessed using wires and catheters under imaging guidance with access typically though the common femoral artery via the groin. This has been shown to be a viable management option in patients who are traumatically injured, in the absence of concurrent immediately life-threatening other injuries requiring damage control surgery (DCS) [1].

Splenic injury is classified according to the American Association of Trauma Surgery grade (grades 1 to 5) [2], with increased severity traumatic injury according to the higher numerical value. SE is typically performed in higher grade (3/4) splenic injuries, although the gold standard of management of Grade 5 is considered surgical resection. There is no current definitive consensus on the management of high grade splenic injuries although there is a trend towards embolisation and splenic conservation since the inception of trauma networks in England in 2012 [3]. The 22 Trauma centres now function as a hub for trauma within their specified area and had the aim of developing trauma services and improving clinical care. The 22 Adult Trauma centres within England are listed in S1 Appendix.

There are few guidelines regarding the availability and design of interventional radiology (IR) provision at Major Trauma Centres (MTCs) [4] and there is no available data on the impact of IR on-call structure and quality or location of IR facilities on the splenic conservation rate and time to treatment. SE technique and rate are variable and depend on multiple factors. These factors include the time to CT report, the availability of on-call IR services, the method of contacting IR, availability of a hybrid theatre and the associated injuries. A recent survey of British Society of Interventional Radiology (BSIR) members, undertaken as part of the BSIR audit and registry committee, demonstrated wide variability in the management and treatment of splenic injuries with respect to SE [5]. This was due to a number of factors regarding service design and decisions around appropriateness and method of embolisation.

SE can be performed in two main ways, either with a proximal occlusion of the main splenic artery outside of its hilum, or within the actual splenic tissue having selected the arterial branch that is demonstrated as bleeding [6]. The embolisation (stopping of the bleeding) can be performed using a variety of embolic agents, including coils, plugs, gelfoam or glue to stop the

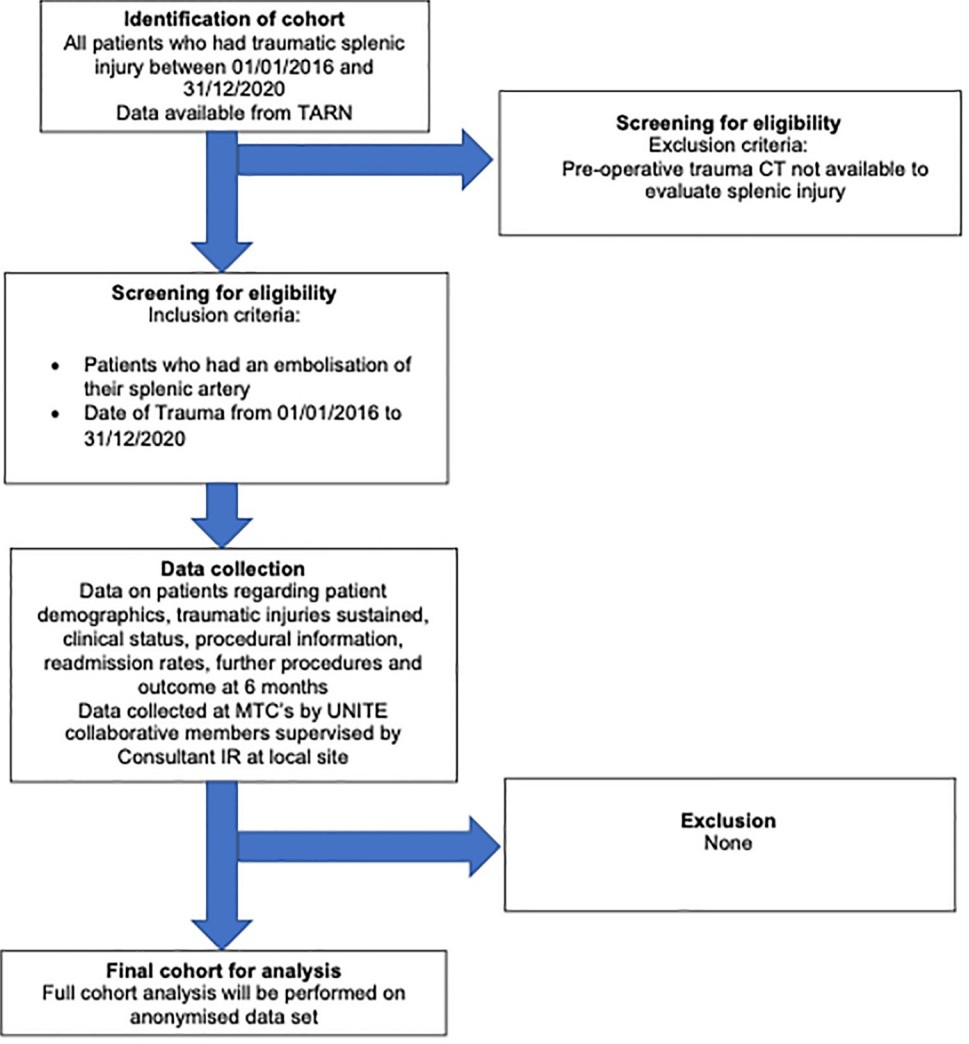

**Fig 1. Study flow chart.**

blood getting to the damaged vessel although the evidence base for which approach and embolic agent remains limited. No multicentre UK based dataset has been published. This highlights the lack of consensus, guidelines, and research in this area. Work on the available retrospective dataset which are available through Trauma and Audit Research Network (TARN) should be undertaken to analyse the current situation to enable design of multi-centre prospective research.

This work will benefit patients by establishing an improved evidence base regarding the optimum service design and treatment pathway. We aim to benefit the NHS by clearly identifying factors that improve the successful embolisation rate, a less invasive procedure than damage control surgery (DCS) whereby a surgeon would remove the spleen through a large incision in the abdomen. We aim to clarify the role of and support the development of IR within the trauma setting by establishing a more evidence-based practice to support interventional radiologists in their decision-making around SE in the context of Acute Traumatic Splenic injury (ATSI). The determination of the impact IR service design on outcomes will enable improved management decisions on overall patient care.

**Table 1. Study summary.**

| Study Title | SPEED (Splenic Embolisation Decisions) |
|---|---|
| Study Design | Multicentre retrospective observational cohort study with corresponding questionnaire. |
| Study Participants | Acute traumatic splenic injury within the Major Trauma Centres in England |
| Eligibility Criteria | Inclusion Criteria:<br>    1. Patients who sustained traumatic splenic injury as identified in the TARN (Trauma Audit and Research Network) database.<br>    2. Date of traumatic injury from 01/01/2016 to 31/12/2020.<br>Exclusion Criteria:<br>    1. CT not available to radiologically grade the Splenic injury |
| Planned Sample Size | 500–600 |
| Follow-up Duration | 2 year from date of trauma or date of death, whichever is sooner |
| Planned Study Period | 12 months |
| Primary Objective | 1. To determine if service design significantly affects splenic embolisation (SE) rates in AAST grade 2–5 acute traumatic splenic injuries (ATSI) across the Major Trauma Centres (MTCs) in the UK.<br>2. To determine if variation in treatment affects SE outcomes in ATSI |
| Secondary Objective | • To identify factors that determine the decision to embolise<br>• To analyse the effect of embolisation agent on outcomes<br>• To compare office hours versus out of hours interventional radiology (IR) trauma service provision across the MTCs and its effect on embolisation rate<br>• To compare embolisation outcomes per volume performed between the MTCs<br>• To evaluate the follow-up imaging pathway and necessity for follow-up imaging in acute splenic injury.<br>• To establish the failure of AAST grade 3 injuries in conservative management.<br>• To establish the rate of re-intervention, embolisation failure and complications.<br>• To evaluate the relationship between AAST grade and outcome<br>• To determine imaging features that are associated with poor outcomes using machine learning. |

## 1.2 Rationale for current study

SE is a treatment option that is variably utilised and has unclear evidence base. We aim to establish the current UK practice and determine how factors (including service design identified by questionnaire) affect splenic artery embolisation usage and outcomes.

## 1.3 Participant and public involvement

Due to the retrospective and observational nature of this study no patient involvement would be appropriate.

## 2. Study objectives
## 2.1 Primary objectives

1. To determine if service design significantly affects splenic embolisation (SE) rates in AAST grade 2–5 acute traumatic splenic injuries (ATSI) across the Major Trauma Centres (MTCs) in the UK.

2. To determine if variation in treatment affects SE outcomes in ATSI

## 2.2 Secondary objectives

- To identify factors that determine the decision to embolise

- To analyse the effect of embolisation agent on outcomes

- To compare office hours versus out of hours interventional radiology (IR) trauma service provision across the MTCs and its effect on embolisation rate

- To compare embolisation outcomes per volume performed between the MTCs

- To evaluate the follow-up imaging pathway and necessity for followup imaging in acute splenic injury.

- To establish the failure of AAST grade 3 injuries in conservative management.

- To establish the rate of re-intervention, embolisation failure and complications

- To evaluate the relationship between AAST grade and outcome

- To determine imaging features that are associated with poor outcomes using machine learning.

- To compare the outcomes of SE by treatment choice

## 2.3 Outcome measures

- SE rate as percentage of; acute splenic trauma, per injury grade and per Major Trauma Centre (MTC). Failure of conservative management rate and splenectomy rate.

- The IR service design and effect on SE rate, e.g service design factors including on call frequency and vacant posts versus embolisation rate.

- Proportion of proximal vs distal embolisation (distal defined by embolisation of a branch distal to the main splenic artery[5]) and as percentage of; ATSI, AAST grade and per MTC.

- SE technical success per technique, splenic salvage rate, major and minor complication rate operative time, readmission rate, further intervention and death

- Splenic surgery rate as percentage of; acute splenic trauma, AAST grade and per MTC

## 3. Study design and methods

The Chief Investigator obtained approval from the Health Research Authority (HRA) and Research Ethics Committee (REC) on 10/04/24 (Approved by HRA and Health and Care Research Wales, REC reference 24/HRA/1596).

We will include 5 years of data from traumatic splenic injury patients in the MTCs from 01/01/2016 to 31/12/2020. This data is currently available from TARN centrally under an already agreed data transfer as audit data with the Sponsor. We estimate MTCs perform around 10 SE per year and approximately 5–10 splenectomies, with one study from 2017 demonstrating rates are increasing to 7.6% of splenic injuries in 2012–2014 [4]. Therefore, across the currently involved 12 MTCs we expect to capture around 500–600 patients who underwent SE over a five-year period. The embolisation rate will be determined as a proportion of the splenic injuries AAST 2–5 documented on the TARN database. The data collection period has been selected to allow for a minimum 2 year follow up.

Data will be available from the TARN database and then correlated with data that will be collected at each MTC, where detail as to the embolisation technique, specific injury pattern, imaging based follow up and patient survival will be available. A short service based questionnaire will be sent to each centre to establish centre-specific systems such as on call rota, IR response activation, reporting practices and capture data around routine decision-making at that site. This will be completed by the site team at registration at each site and will take approximately 5 minutes to complete. This will feed into the assessment of the service design to allow analysis of the impact of these service design features.

Data will be collected by a named radiology registrar as part of the IR trainee research collaborative (UNITE) at each participating MTC and will be maintained on an anonymised (REDCap) database. Data collection and imaging review is estimated to take no longer than 15 minutes per case, with most hospitals having less than 30 patients. Registrars will collect this data as part of their training time, where there is a requirement to be involved in audit and research. No patients will be involved in this study and all data will be anonymised prior to leaving the MTC.

Step by step methods:

1. TARN Data provided to Chief investigator (this is available under an Audit agreement already)

2. A general questionnaire will be completed by the site team at sit registration to provide general site-specific information via secure email.

3. Data will be sorted by SITE identification code available in TARN dataset

4. Individual sites registrars, recruited under the UNITE interventional radiology trainee research group will be provided with a list of TARN identification numbers, date of injury and first CT report at their site via secure email.

5. Individual site registrars will use these three factors to identify the imaging within the radiological identification system at the local hospital

6. Additional data will then be collected and anonymously inputted by individual site registrars into the REDcap database using the TARN identifiable number only

7. The additional anonymous data from REDCap will then be merged with the available TARN dataset using the TARN identification number by the CI

8. At no point will the data be identifiable outside of the site at which the patient was injured

## 4. Study participants

### 4.1 Screening procedures

All patients identified as having an acute traumatic splenic injury grade AAST 2–5 on the TARN database between the given dates will be included. Hospitals who decline participation in the data collection will have their patients excluded from the analysis excluded by site as identified in TARN database

### 4.2 Inclusion criteria

All patients who had traumatic splenic injury between 01/01/2016 and 31/12/2020 with data available from TARN CT available for review.

### 4.3 Exclusion criteria

CT not available to radiologically grade the Splenic injury

## 5. Study procedures and interventions

### 5.1 Recruitment

The retrospective observational data will be obtained from TARN. No patient specific recruitment will be required.

### 5.2 Consent

Consent will not be required due to the retrospective and anonymous nature of the data. There will be no change in the patients' treatment pathway. Consent for the questionnaire is not appropriate due to its general nature about the registered site, rather than the person answering.

### 5.3 Definition of end of study

This is defined as the date of the last data submitted by the site. The sponsor will notify the REC, in writing, within 90 days of the end of the study

## 6. Statistics

### 6.1 The number of participants

We estimate MTCs perform around 10 SE per year and approximately 5–10 splenectomies, with one study from 2017 demonstrating rates are increasing to 7.6% of splenic injuries in 2012–2014.[4] Therefore, across the involved MTCs we expect to capture around 500–600 patients who underwent SE over a five-year period

### 6.2 Analysis of endpoints

Data will be analysed using SPSS v27 (IBM Corp. Armonk, NY). Statistic support from the University of Plymouth statistical department has been obtained.

Logistical regression analysis will be used to identify factors affecting the embolisation rates, presented with odds ratios. A volume outcome analysis performed to assess the impact of caseload on outcomes. Categorical data will be assessed within Chi squared test with continuous data will likely be assess with Mann-Whitney U test, depending on normality assessment. P values of $<0.05$ will be taken as significant. Survival analysis will be examine the data using a Kaplan-Meier curve.

### 6.3 Data flow diagram

The SPEED data flow chart is shown in Fig 2.

### 6.4 Description of the data

The study data points are shown in Table 2.

### 6.5 Collection of data and study materials

- TARN Database–Held on secure Trust computer

- REDCap database

- Questionnaire–held on secure online format *via* secure email.

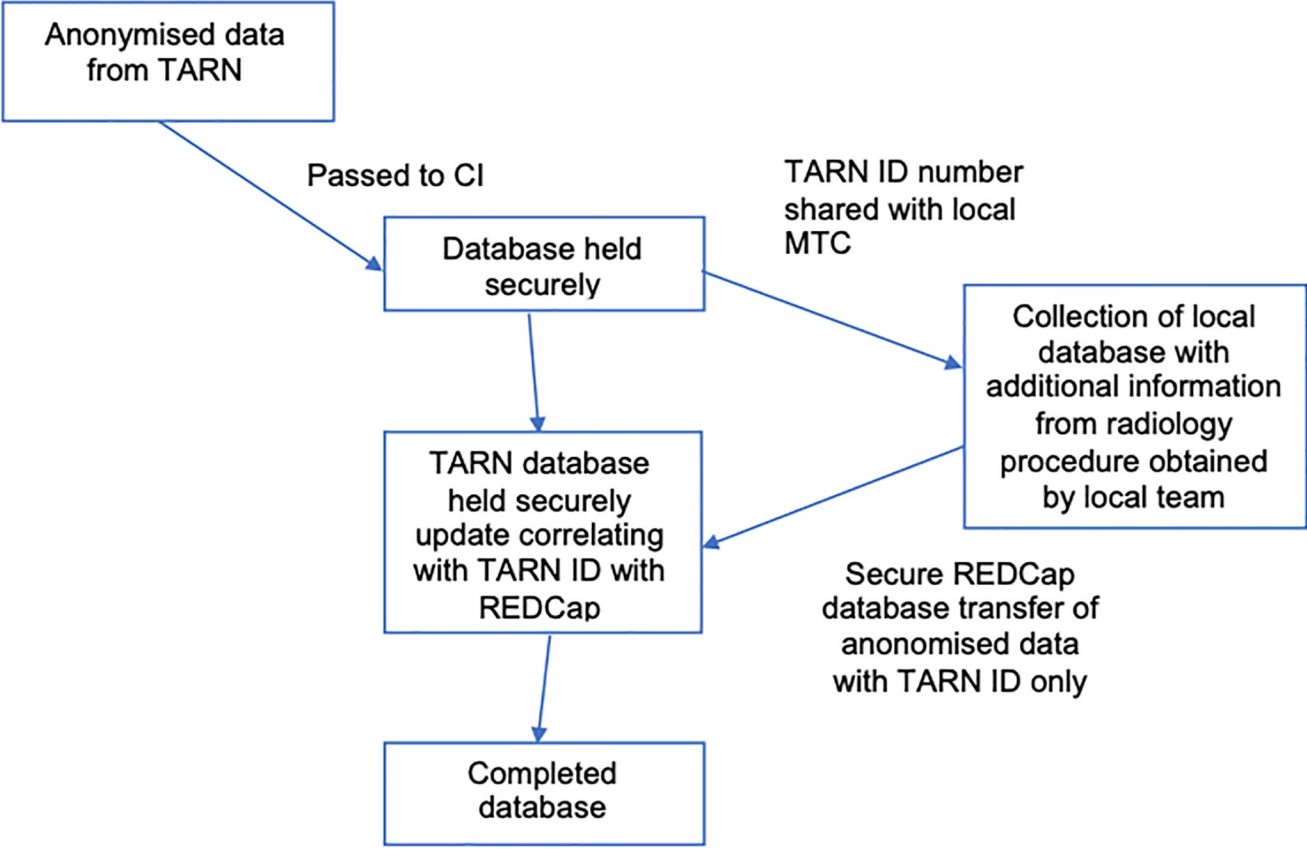

**Fig 2. SPEED data flow chart.** Abbreviations: CI = Chief Investigator, TARN = trauma audit research network. MTC = major trauma centre.

## 6.6 Data storage and security

- TARN data will be sent via secure email and stored on a Trust computer in a password protected file. Data will be transferred in anonymised fashion from TARN to the Chief Investigator *via* secure email and stored on a Trust computer. The data from TARN is coded by a TARN ID number which can be only linked back to the patient by the site. However, neither the CI team nor sponsor will be able to link any data back to the patient, therefore is considered anonymous data

- All anonymous patient data will be stored on REDCap maintained by the CI with oversight from the sponsor.

- The questionnaire will be completed by the site team at each site via a secure electronic online form system, where the data will be stored.

- Questionnaire data will be stored on a Trust computer in a password protected file with access only available by the CI and shared with the statistics team

## 6.7 Archiving, preservation and curation

Archiving will be authorised by the Sponsor following submission of the end of study declaration. Upon completion of the study, study documents will be archived for a minimum of 5

**Table 2. SPEED data variables to be collected.**

| TARN | | MTC Radiological information system, PACS and Clinical systems | Site registration Questionnaire |
|---|---|---|---|
| • Trauma Time (mm:hh)<br>• Mechanism of Injury<br>• Glasgow coma scale (GCS)<br>• Intubation (Y/N)<br>• P Score<br>• Observations<br>• Tranexamic acid (Y/N)<br>• Trauma Call (Y/N)<br>• Blood products received (Y/N)<br>• Massive Transfusion protocol activation (Y/N)<br>• IR attend trauma call (Y/N)<br>• Injuries sustained<br>• Time to CT scan<br>• Time from CT scan to radiology report<br>• Injury Severity Score (ISS)<br>• AAST grade<br>• Embolisation (Y/N)<br>• Anticoagulation (Y/N) | • Injuries sustained<br>• Time to CT scan<br>• Time from CT scan to radiology report<br>• Injury Severity Score (ISS)<br>• AAST grade<br>• Embolisation (Y/N)<br>• Anticoagulation (Y/N) | • Time of discussion with IR<br>• Splenic trauma grade (AAST)<br>• CT appearances including location and descriptors<br>• Digital subtraction angiography (Y/N)<br>• Embolisation (Y/N)<br>• Hybrid approach (Y/N)<br>• Embolization agent<br>• Time to intervention, procedure length<br>• Technical success (Y/N)<br>• Required operative management (Y/N)<br>• Time to rebleed<br>• Discharge destination<br>• Survival to discharge/ 30 mortality<br>• Antibiotic usage (Y/N)<br>• Follow up imaging to 2 years (Y/N)<br>• Rebleed (Y/N)<br>• Failure of conservative management (Y/N) | Number of Consultants at MTC<br>• Population served by MTC<br>• Number of trauma units in network<br>• Number of IR rooms<br>• Hybrid theatre (Y/N)<br>• IR nursing on call pattern<br>• Stroke thrombectomy sharing on call team?<br>• IR attend Trauma calls<br>• Process of On-call alert for IR<br>• Diagnostic Trauma Reporting set up |

Abbreviations: AAST = American Association for the Surgery of Trauma, GCS = Glasgow coma scale, ISS = Injury Severity Score, MTC = major trauma centre.

years as per the participating Trust's Research Archiving SOP. Once the archiving retention period has been reached, the Sponsor will liaise with the CI regarding destruction.

## 7. Study management

The day-to-day management of the study will be co-ordinated through Dr Paul Jenkins.

## 8. Publication policy

Final results of the study will be disseminated *via* presentations at appropriate scientific meetings, including the BSIR Annual conference and publication in appropriate peer-reviewed journals. Authorship will involve named individuals involved in study design and manuscript preparation in conjunction with the British Society of Interventional Radiology Trainee Research group, with individuals collecting data at hospital sites being named as collaborators.

## Supporting information

**S1 Appendix.**
(DOCX)

## Author Contributions

**Conceptualization:** Paul Jenkins, Jim Zhong, James Harding, Jason Smith, Carl Roobottom.

**Data curation:** Paul Jenkins, Lexy Sorrell, Jason Smith, Victoria Allgar.

**Formal analysis:** Paul Jenkins, Jim Zhong, Lexy Sorrell, Victoria Allgar.

**Funding acquisition:** Paul Jenkins, Jim Zhong, Jason Smith, Carl Roobottom.

**Investigation:** Paul Jenkins, Lexy Sorrell, Jason Smith, Carl Roobottom.

**Methodology:** Paul Jenkins, Jim Zhong, Lexy Sorrell, Jason Smith, Victoria Allgar, Carl Roobottom.

**Project administration:** Paul Jenkins, Jim Zhong, Jason Smith.

**Resources:** Paul Jenkins, Lexy Sorrell.

**Software:** Paul Jenkins, Victoria Allgar.

**Supervision:** Paul Jenkins, Jim Zhong, James Harding, Victoria Allgar, Carl Roobottom.

**Validation:** Paul Jenkins, James Harding, Lexy Sorrell, Victoria Allgar.

**Visualization:** Paul Jenkins, Lexy Sorrell, Victoria Allgar.

**Writing – original draft:** Paul Jenkins, Jim Zhong.

**Writing – review & editing:** Paul Jenkins, Jim Zhong, James Harding, Jason Smith, Carl Roobottom.

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
