## [Decision Letter · Decision Letter 0]

1 Oct 2024

PONE-D-24-31640SPEED (SPlEnic Embolisation Decisions) Study – Decision to treat acute traumatic splenic artery injury in the context of trauma protocolPLOS ONE

Dear Dr. Zhong,

Thank you for submitting your manuscript to PLOS ONE. After careful consideration, we feel that it has merit but does not fully meet PLOS ONE’s publication criteria as it currently stands. Therefore, we invite you to submit a revised version of the manuscript that addresses the points raised during the review process.

We look forward to receiving your revised manuscript.

Kind regards,

Eyüp Serhat Çalık

Academic Editor

PLOS ONE

Journal Requirements:

1. When submitting your revision, we need you to address these additional requirements. Please ensure that your manuscript meets PLOS ONE's style requirements, including those for file naming. The PLOS ONE style templates can be found at https://journals.plos.org/plosone/s/file?id=wjVg/PLOSOne_formatting_sample_main_body.pdf and https://journals.plos.org/plosone/s/file?id=ba62/PLOSOne_formatting_sample_title_authors_affiliations.pdf 2. Thank you for stating the following financial disclosure: "British Society of Interventional Radiology Research Grant " Please state what role the funders took in the study.  If the funders had no role, please state: "The funders had no role in study design, data collection and analysis, decision to publish, or preparation of the manuscript." If this statement is not correct you must amend it as needed. Please include this amended Role of Funder statement in your cover letter; we will change the online submission form on your behalf. 3. When completing the data availability statement of the submission form, you indicated that you will make your data available on acceptance. We strongly recommend all authors decide on a data sharing plan before acceptance, as the process can be lengthy and hold up publication timelines. Please note that, though access restrictions are acceptable now, your entire data will need to be made freely accessible if your manuscript is accepted for publication. This policy applies to all data except where public deposition would breach compliance with the protocol approved by your research ethics board. If you are unable to adhere to our open data policy, please kindly revise your statement to explain your reasoning and we will seek the editor's input on an exemption. Please be assured that, once you have provided your new statement, the assessment of your exemption will not hold up the peer review process.

Additional Editor Comments:

I would like to thank the authors for this important study they have planned. I think that the study protocol is generally well-written, and when completed, it will fill an important gap in the field. However, I have a few minor editing suggestions.

- There is no standard for abbreviations throughout the manuscript; if possible, abbreviations should not be used in the abstract. In the main text, the abbreviation should be written clearly at the first mention, the abbreviation should be defined in parentheses and the abbreviation should be used in all subsequent parts of the text.

- Are there any hybrid approaches in the centers where the study is planned? For example, is embolization performed first, followed by laparoscopic or small laparotomy incision to drain the hematoma or remove the spleen? If so, I think such hybrid cases should also be included in the protocol.

- Why didn't you consider comparing splenic embolization with open surgery when evaluating the outcomes of splenic embolization, which you said the open surgery is the gold standard in the menagement of splenic injury especially in grade 5 patients.

- Why did you limit your follow-up data to 6 months?

Good luck.

Reviewers' comments:

Reviewer's Responses to Questions

**Comments to the Author**

1. Does the manuscript provide a valid rationale for the proposed study, with clearly identified and justified research questions?

Reviewer #1: Yes

Reviewer #2: No

2. Is the protocol technically sound and planned in a manner that will lead to a meaningful outcome and allow testing the stated hypotheses?

Reviewer #1: Yes

Reviewer #2: Partly

3. Is the methodology feasible and described in sufficient detail to allow the work to be replicable?

Reviewer #1: Yes

Reviewer #2: No

4. Have the authors described where all data underlying the findings will be made available when the study is complete?

Reviewer #1: Yes

Reviewer #2: No

5. Is the manuscript presented in an intelligible fashion and written in standard English?

Reviewer #1: Yes

Reviewer #2: No

6. Review Comments to the Author

You may also provide optional suggestions and comments to authors that they might find helpful in planning their study.

Reviewer #1: The protocol is well written encompassing all the relevant details. t will be interesting to see the outcome of this study once completed.

Reviewer #2: please revise the paper.

please revise the paper.

please revise the paper.

please revise the paper.

please revise the paper.

7. PLOS authors have the option to publish the peer review history of their article (what does this mean?). If published, this will include your full peer review and any attached files.

Reviewer #1: No

Reviewer #2: **Yes: **hazim abdul rahman alhiti

---

## [Author Response · Author response to Decision Letter 0]

14 Oct 2024

Thank you for your comments.

As per the review, the funding disclosure statement has been updated with: 

- There is no standard for abbreviations throughout the manuscript; if possible, abbreviations should not be used in the abstract. In the main text, the abbreviation should be written clearly at the first mention, the abbreviation should be defined in parentheses and the abbreviation should be used in all subsequent parts of the text.

This has been altered. 

- Are there any hybrid approaches in the centers where the study is planned? For example, is embolization performed first, followed by laparoscopic or small laparotomy incision to drain the hematoma or remove the spleen? If so, I think such hybrid cases should also be included in the protocol.

This has been clarified to ensure this is included in the data collection.

- Why didn't you consider comparing splenic embolization with open surgery when evaluating the outcomes of splenic embolization, which you said the open surgery is the gold standard in the management of splenic injury especially in grade 5 patients.

Thank you for your suggestion – This has now been included

- Why did you limit your follow-up data to 6 months?

Thankyou for your comment – This has now been extended to 2 years.

---

## [Editor Report · Decision Letter 1]

21 Oct 2024

SPEED (SPlEnic Embolisation Decisions) Study – Decision to treat acute traumatic splenic artery injury in the context of trauma protocol

PONE-D-24-31640R1

Dear Dr. Zhong,

We’re pleased to inform you that your manuscript has been judged scientifically suitable for publication and will be formally accepted for publication once it meets all outstanding technical requirements.

Kind regards,

Eyüp Serhat Çalık

Academic Editor

PLOS ONE

Additional Editor Comments (optional):

I would like to thank the authors for their revisions and appropriate responses.
---

## [Editor Report · Acceptance letter]

5 Nov 2024

PONE-D-24-31640R1 

PLOS ONE

Dear Dr. Zhong, 

I'm pleased to inform you that your manuscript has been deemed suitable for publication in PLOS ONE. Congratulations! Your manuscript is now being handed over to our production team.

Kind regards, 

on behalf of

Dr. Eyüp Serhat Çalık 

Academic Editor

PLOS ONE